# Temperature sensitivity of the interspecific interaction strength of coastal marine fish communities

**Masayuki Ushio[1,2,3]\*, Testuya Sado[4], Takehiko Fukuchi[4], Sachia Sasano[5,6], Reiji Masuda[5], Yutaka Osada[7], Masaki Miya[4]\***

[1]Hakubi Center, Kyoto University, Kyoto, Japan; [2]Center for Ecological Research, Kyoto University, Otsu, Japan; [3]Department of Ocean Science, The Hong Kong University of Science and Technology, Clear Water Bay, Kowloon, Hong Kong SAR, China; [4]Natural History Museum and Institute, Chiba, Japan; [5]Maizuru Fisheries Research Station, Kyoto University, Maizuru, Japan; [6]Fisheries Technology Institute, Japan Fisheries Research and Education Agency, Ishigaki, Japan; [7]Graduate School of Life Sciences, Tohoku University, Sendai, Japan

## eLife assessment

This study presents **important** findings regarding the quantification of dynamics in fish communities in changing ecosystems by combining a large-scale environmental DNA metabarcoding time series with novel statistical approaches. The methods are **convincing**, with controlled experiments, thorough statistical analyses, and a substantial dataset covering two years of detailed observation, which can provide sufficient power to detect fine-scale ecological interactions. This work is relevant for informing future research on assessing community stability under climate change.

**\*For correspondence:**
ong8181@gmail.com (MU);
masaki_miya@me.com (MM)

**Competing interest:** The authors declare that no competing interests exist.

**Abstract** The effects of temperature on interaction strengths are important for understanding and forecasting how global climate change impacts marine ecosystems; however, tracking and quantifying interactions of marine fish species are practically difficult especially under field conditions, and thus, how temperature influences their interaction strengths under field conditions remains poorly understood. We herein performed quantitative fish environmental DNA (eDNA) metabarcoding on 550 seawater samples that were collected twice a month from 11 coastal sites for 2 years in the Boso Peninsula, Japan, and analyzed eDNA monitoring data using nonlinear time series analytical tools. We detected fish–fish interactions as information flow between eDNA time series, reconstructed interaction networks for the top 50 frequently detected species, and quantified pairwise, fluctuating interaction strengths. Although there was a large variation, water temperature influenced fish–fish interaction strengths. The impact of water temperature on interspecific interaction strengths varied among fish species, suggesting that fish species identity influences the temperature effects on interactions. For example, interaction strengths that *Halichoeres tenuispinis* and *Microcanthus strigatus* received strongly increased with water temperature, while those of *Engraulis japonicus* and *Girella punctata* decreased with water temperature. An increase in water temperature induced by global climate change may change fish interactions in a complex way, which consequently influences marine community dynamics and stability. Our research demonstrates a practical research framework to study the effects of environmental variables on interaction strengths of marine communities in nature, which would contribute to understanding and predicting natural marine ecosystem dynamics.

**eLife digest** The world's oceans are home to tens of thousands of fish species, many of which live in nutrient-rich coastal waters. Different species living in a particular environment interact with each other in many ways. For example, a predatory fish may prey on some species of small fish but avoid feeding on others that help it by removing parasites from its skin. Rising ocean temperatures caused by global climate change could affect how different fish species interact with one another and, as a result, impact their communities.

One of the first steps to understanding how fish interact with each other in nature typically requires researchers to count the number of different species present and observe how they behave, which is time-consuming and labor-intensive. An alternative is to use an emerging technique in which researchers extract DNA from water, soil or air – known as environmental DNA – and analyze it to identify the species present and estimate their numbers.

Ushio et al. analyzed hundreds of samples of seawater that had been collected over a two-year period from the Boso Peninsula in Japan. Statistical methods were used to quantify how strongly fish species interact with each other and determine whether the temperature of the water influenced how different species of fish interacted over time. The findings showed that water temperature had a significant but complex effect on how strongly pairs of fish species interacted, with both positive and negative effects depending on the conditions. The impact of water temperature on the strength of the interactions varied between species, for example, Japanese anchovy and largescale blackfish interacted less strongly with other fish species in warmer water, whereas the Stripey and a species of wrasse interacted with other fish species more strongly.

The findings provide new insights into how water temperature affects the communities of fish living in coastal areas. Alongside complementing existing knowledge in the field, refining the research framework used in this work will benefit those working in fishery science by providing valuable insights into how natural and commercially important fish species respond to climate change.

## Introduction

Interspecific interactions are key to understanding and predicting the dynamics of ecological communities (*May, 1972*; *Tang et al., 2014*; *Wootton and Emmerson, 2005*; *Wootton and Stouffer, 2016*). Theoretical and empirical studies have shown that various properties of interspecific interactions, namely, the number of interactions, sign (positive or negative), strength, and correlations of pairwise interactions (e.g., predator–prey interactions), influence the dynamics, stability, and diversity of ecological communities in terrestrial and aquatic ecosystems (*Allesina et al., 2015*; *Mougi and Kondoh, 2012*; *Ratzke et al., 2020*; *Tang et al., 2014*; *Ushio et al., 2018a*; *Ushio, 2022a*). Interaction strength is a fundamental property, and previous studies examined the relationship between interaction strengths and ecological community properties (*Wootton and Emmerson, 2005*; *Wootton and Stouffer, 2016*). The dominance of weak interactions stabilizes the dynamics of a natural fish community (*Ushio et al., 2018a*). Interaction strengths have been reported to decrease with increases in the diversity of experimental microbial communities (*Ratzke et al., 2020*), and this holds true even for more diverse ecological communities under field conditions (*Ushio, 2022a*).

Environmental variables exert significant effects on interspecific interaction strengths. The effects of temperature on interaction strengths are important for understanding and forecasting the impact of the ongoing global climate change on ecosystems. The relationship between temperature and interaction strengths has been investigated in terrestrial and aquatic ecosystems for decades (*Adams and Zhang, 2009*; *Allan et al., 2015*; *Coley and Aide, 1991*; *Hein et al., 2014*; *Kishi et al., 2005*; *Kordas et al., 2011*; *Kratina et al., 2012*; *Rall et al., 2010*; *Thakur et al., 2017*). In terrestrial ecosystems, *Coley and Aide, 1991* proposed that interactions between plants and insect herbivores were generally stronger in warmer regions, which may result in stronger negative density dependence for tree species and contribute to higher plant diversity in a tropical region (*Forrister et al., 2019*). The influence of temperature on interaction strengths has also been investigated in other systems: terrestrial arthropods interactions (*Rall et al., 2010*), fish–fish interactions (*Allan et al., 2015*; *Hein et al., 2014*), and fish–prey interactions (*Kishi et al., 2005*). In addition, *Wieczynski et al., 2021* showed that species-level functional traits (e.g., body size and shape) may underlie the relationships

between temperature and interaction strengths. Therefore, interplays among temperature, interaction strengths, species identity, and community-level function (e.g., primary production) are prevalent, and a more detailed understanding of the complex interactions among them is critical for predicting ecosystem responses to climate change.

In marine ecosystems, interspecific interactions in ecological communities play a fundamental role in system dynamics, such as population dynamics, primary production, and nutrient cycling (*Hannisdal et al., 2017*; *Penn et al., 2019*; *Ushio et al., 2018a*), as well as ecosystem services, including food supply (*Smith et al., 2019*). Although many studies have evaluated the effects of temperature on interaction strengths of marine organisms (*Kordas et al., 2011*), most of these studies have been performed targeting relatively immobile or small organisms (*Bertness and Ewanchuk, 2002*; *Chen et al., 2012*) or under laboratory conditions (e.g., mesocosm experiments; *Allan et al., 2015*). While the previous studies have provided invaluable information, it currently remains unclear how temperature influences the strengths of interspecific interactions of larger, more mobile marine organisms, such as fish, which may exert strong top-down regulations on ecological communities, especially under field condition. This may be due to the difficulties associated with detecting and measuring interaction strengths among multiple, relatively large, mobile species under field conditions; the identification of quickly moving fish species and the quantification of their abundance under field conditions are challenging, and the quantification of their interactions is even more difficult. Overcoming these difficulties and understanding how temperature influences the strength of interspecific interactions of marine fish communities will provide insights into how marine fish communities are assembled, how they may respond to the ongoing and future global climate change, and how the changes in fish communities may transmit to other trophic levels.

Environmental DNA (eDNA), defined here as extra-organismal DNA left behind by macro-organisms (*Bohmann et al., 2014*), has been attracting increasing attention as an indirect genetic marker for inferring the presence of species for biodiversity monitoring (*Cristescu and Hebert, 2018*; *Deiner et al., 2017*). A simple protocol for collecting eDNA samples from aquatic environments facilitates continuous biodiversity monitoring at multiple sites (*Deiner et al., 2017*), and eDNA metabarcoding (the simultaneous detection of multiple species using universal primers and a high-throughput sequencer) provides useful information on the dynamics of ecological communities (*Bálint et al., 2018*; *Miya et al., 2020a*; *Miya, 2022*; *Ushio, 2022a*). Recent studies demonstrated that eDNA metabarcoding combined with frequent water sampling enabled the efficient monitoring of high-diversity ecological communities (*Bista et al., 2017*; *Djurhuus et al., 2020*; *Ushio, 2022a*). Furthermore, if eDNA metabarcoding is performed quantitatively (e.g., by including spike-in DNAs; *Ushio et al., 2018b*), these data may contain information on species abundance.

More importantly, recent studies have shown that information on interspecific interactions may be embedded in multispecies eDNA time series, particularly when the data are 'quantitative' (*Ushio, 2022a*). Advances in nonlinear time series analyses have enabled the quantification of interspecific interactions only from time series data. For example, transfer entropy (TE) is a method based on the information theory that quantifies information flow between two variables (*Runge et al., 2012*; *Schreiber, 2000*). Information flow can be an index of interspecific interactions when applied to ecological data. Convergent cross mapping (CCM) is a causality detection tool in empirical dynamic modeling (*Sugihara et al., 2012*), which is based on the dynamical theory. TE and CCM have more recently been understood under the information theory framework, and unified information-theoretic causality (UIC) may also quantify information flow between variables (*Osada et al., 2023*). In addition, an improved version of a sequentially weighted global linear map (S-map), called the multiview distance regularized (MDR) S-map, enables accurate quantifications of interaction strengths of a large interaction network even when the number of network nodes exceeds the time series length (*Chang et al., 2021*). These advanced statistical tools facilitate the detection and quantification of interspecific interactions from quantitative, multispecies eDNA time series data.

In the present study, we collected eDNA samples twice a month from 11 coastal sites at the southern tip of the Boso Peninsula, central Japan (*Figure 1a*) for 2 years (a total of 550 samples). This region is located on the Pacific side of Honshu Island around 35°N and is markedly affected by the warm Kuroshio Current, cold Oyashio Current, and inner-bay water from Tokyo Bay. These geographic and oceanographic characteristics form latitudinal temperature gradients, allowing us to investigate temperature effects on interspecific interactions. We obtained quantitative eDNA metabarcoding

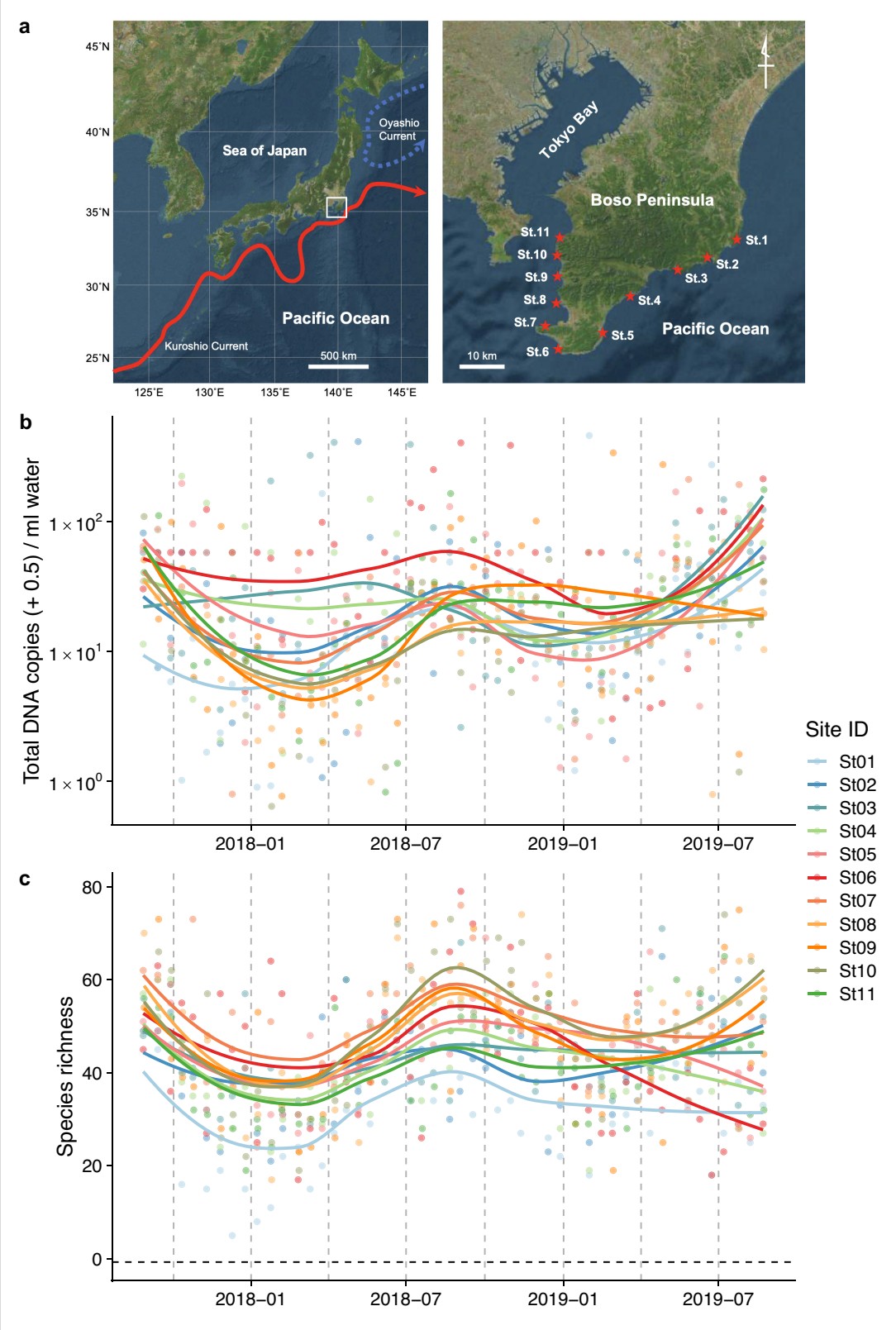

**Figure 1.** Study sites and overall dynamics of environmental DNA (eDNA) concentrations and the number of fish species detected. (**a**) Study sites in the Boso Peninsula. The study sites are influenced by the Kuroshio Current (red arrow; left panel) and distributed along the coastal line in the Boso Peninsula (right panel). (**b**) Total eDNA copy numbers estimated by quantitative eDNA metabarcoding (see Methods for detail). (**c**) Fish species richness detected

*Figure 1 continued on next page*

*Figure 1 continued*

by eDNA metabarcoding. Points and lines indicate raw values and LOESS lines, respectively. The line color indicates the sampling site. Warmer colors generally correspond to study sites with a higher mean water temperature.

The online version of this article includes the following figure supplement(s) for figure 1:

**Figure supplement 1.** Dynamics of environmental DNA (eDNA) copy numbers of Japanese black seabream (*Acanthopagrus schlegelii*) that was used as an internal standard.

**Figure supplement 2.** Dynamics of water temperature and the relationships between water temperature and total environmental DNA (eDNA) concentration and fish species richness.

data by combining eDNA metabarcoding data (relative abundance data) and the eDNA concentration data obtained through quantitative PCR (qPCR) for one of the most common fish species, *Acanthopagrus schlegelii* (e.g., absolute abundance data of *A. schlegelii* eDNA were utilized as an internal spike-in DNA; see Methods). Based on quantitative, multispecies eDNA time series data, pairwise species interactions were detected as information flow between species using a nonlinear time series analysis (*Osada et al., 2023*). In addition, interaction strengths at each time point were quantified for the detected fish–fish interactions using the MDR S-map method (*Chang et al., 2021*). As our eDNA time series was taken twice a month, the interactions detected should also have the same time scale (e.g., the interactions detected may cause changes in the population size at the same time scale), which means that we tend to focus on behavior-level interactions (e.g., feeding, vigilance, and schooling) rather than birth–death process in the present study (except for predation). In addition, the interactions we detected by the time series analysis could include various types of interactions such as competition and mutualism that could be involved in the population dynamics. We hypothesized that (1) a positive relationship exists between temperature and interaction strengths at the community level, as found in other ecosystems and many types of interactions, and (2) the relationship between the patterns of temperature and interaction strengths varies among fish species because of species-specific ecologies and the behavior of component species.

## Materials and methods
### Study sites and environmental observations

We selected 11 sites (Stations [Sts.] 1–11) for seawater sampling along the shoreline of the southern tip of the Boso Peninsula (*Figure 1a*). The 11 stations are located within or surrounded by rocky shores with intricate coastlines. They are arranged such that the five stations on the Pacific and Tokyo Bay sides are approximately at the same latitudinal intervals, with St. 6 in between. The northernmost stations on both sides (Sts. 1 and 11) are markedly affected by the cold Oyashio Current (*Yang et al., 1993*) and inner-bay water from Tokyo Bay (*Fujiwara and Yamada, 2002*), respectively, while the remaining stations (Sts. 2–10) are markedly affected by the warm Kuroshio Current flowing northward along the Pacific coast as well as its branches (*Soh, 2003*). Before seawater sampling at each station, we measured seawater temperature (°C) and salinity (‰) using a portable water-quality meter (WQC-30, Toa DKK, Tokyo, Japan). In addition, we recorded the sampling start time, latitude/longitude, weather and sea conditions, and turbidity.

### Collection of eDNA samples

We collected seawater samples twice a month (i.e., around the first and second quarter moons) for 2 years between August 2017 and August 2019 at the 11 sites. We employed low-tech bucket sampling to collect seawater using a folding 7.8 l polypropylene bucket (Soft Bucket 8, ISETO, Osaka, Japan) fastened to a 15-m rope (vinylon rope, φ6 mm). Prior to seawater sampling, we wore disposable gloves on both hands and assembled a set of on-site filtration kits consisting of a Sterivex filter cartridge (pore size of 0.45 μm; Merck Millipore, MA, USA) and a 50-ml disposable syringe with a Luer lock connector (TERUMO, Tokyo, Japan). We then thoroughly decontaminated the bucket with a foam-style 10% bleach solution and brought the equipment to the sampling site. We fixed the end of the 15 m rope fastened to the bucket and collected surface seawater by casting and retrieving the bucket full of seawater. We repeated this collection of seawater 10 times to minimize sampling biases at each station.

Two researchers performed on-site filtration using two pairs of the above kit (filter cartridge + syringe) to obtain duplicate samples. With each collection of seawater, we removed the filter cartridge from the syringe, drew approximately 50 ml seawater into the syringe by pulling the plunger, reattached the filter cartridge to the syringe, and pushed the plunger for the filtration of seawater. We repeated this step twice in a single cast of the bucket, and the final filtration volume reached 1000 ml × 2 with ten casts of the bucket. When the filter was clogged before reaching 1000 ml filtration, we recorded the total volume of water filtered. After on-site filtration, we added 1.6 ml of RNAlater (Thermo Fisher Scientific, DE, USA) to the cartridge to prevent eDNA degradation. We made a filtration blank (FB) by filtering 500 ml of purified water in the same manner at the end of each day of water sampling. We transported the filtered cartridges to the laboratory in a portable cooler with ice packs and kept these cartridges at −20°C in the freezer until eDNA extraction.

## eDNA extraction

We thoroughly sterilized the workspace and equipment before DNA extraction. We used filtered pipette tips and conducted all eDNA extraction manipulations in a dedicated room that was separate from the pre- and post-PCR rooms to safeguard against cross-contamination from PCR products.

We extracted eDNA from the filter cartridges using the DNeasy Blood & Tissue kit (QIAGEN, Hilden, Germany) following the methods developed and visualized by *Miya et al., 2016* with slight modifications (*Minamoto et al., 2021*; *Miya and Sado, 2019*). Briefly, we connected the inlet port of each filter cartridge to a 2.0-ml collection tube and tightly sealed the connection between the cartridge and collection tube with Parafilm. We inserted the combined unit into a 15-ml conical tube and centrifuged the capped conical tube at 6000 × *g* for 1 min to remove redundant seawater and RNAlater. After centrifugation, we discarded the collection tube and used an aspirator (QIAvac 24 Plus, QIAGEN, Hilden, Germany) to completely remove any liquid remaining in the cartridge.

We subjected the filter cartridge to lysis using proteinase K. Before lysis, we mixed PBS (220 μl), proteinase K (20 μl), and buffer AL (200 μl), and gently pipetted the mixed solution into the cartridge

**Table 1.** Primer sequences used in the present study.

| Primer information | Primer sequence (5′– 3′)*,†,‡, §,¶ | Length |
|---|---|---|
| *1st PCR primers* | | |
| MiFish-U-forward | *ACACTCTTTCCCTACACGACGCTCTTCCGATCT* NNNNNN GTCGGTAA AACTCGTGCCAGC | 60 |
| MiFish-U-reverse | *GTGACTGGAGTTCAGACGTGTGCTCTTCCGATCT* NNNNNN CATA GTGGGGTATCTAATCCCAGTTTG | 67 |
| MiFish-E-forward-v2 | *ACACTCTTTCCCTACACGACGCTCTTCCGATCT* NNNNNN RGTTGGTAAATCTCGTGCCAGC | 61 |
| MiFish-E-reverse-v2 | *GTGACTGGAGTTCAGACGTGTGCTCTTCCGATCT* NNNNNN GCAT AGTGGGGTATCTAATCCTAGTTTG | 68 |
| MiFish-U2-forward | *ACACTCTTTCCCTACACGACGCTCTTCCGATCT* NNNNNN GCCGGTAA AACTCGTGCC | 57 |
| MiFish-U2-reverse | *GTGACTGGAGTTCAGACGTGTGCTCTTCCGATCT* NNNNNN CATA GGAGGGTGTCTAATCCCCGTTTG | 67 |
| *2nd PCR primers* | | |
| 2nd-PCR-forward | AATGATACGGCGACCACCGAGATCTACAC XXXXXXXX ACACTCTT TCCCTACACGACGCTCTTCCGATCT | 70 |
| 2nd-PCR-reverse | CAAGCAGAAGACGGCATACGAGAT XXXXXXXX GTGACTGGAGTT CAGACGTGTGCTCTTCCGATCT | 66 |

*Normal characters indicate target-specific universal primers (i.e., MiFish primers).
†The six random bases (Ns) in the middle of the 1st PCR primers were appended to enhance cluster separation on flow cells.
‡Italic characters indicate Illumina sequencing primers.
§X indicates index sequences to identify each sample.
¶Underlined characters indicate P5/P7 adapter sequences for Illumina sequencing.

and incubated the cartridge at 56°C for 20 min while stirring the cartridge using a rotator (10 rpm; Mini Rotator ACR-100, AS ONE, Tokyo, Japan). After the incubation, we collected the lysate and purified the DNA extract (ca. 900 µl) using the DNeasy Blood and Tissue kit following the manufacturer's protocol and set the final elution volume at 200 µl. We also made an extraction blank (EB) during this process in addition to FB.

## Paired-end library preparation and MiSeq sequencing

We thoroughly sterilized the workspace and equipment in the pre-PCR area before library preparation. We used filtered pipette tips while performing pre- and post-PCR manipulations in two different dedicated rooms to safeguard against cross-contamination.

We employed two-step PCR for paired-end library preparation using the MiSeq platform (Illumina, CA, USA). We generally followed the methods developed by *Miya et al., 2015* and subsequently modified by *Miya and Sado, 2019*. In the first-round PCR (1st PCR), we used a mixture of the following six primers: MiFish-U-forward, MiFish-U-reverse, MiFish-E-forward-v2, MiFish-E-reverse-v2, MiFish-U2-forward, and MiFish-U2-reverse (*Table 1*). These primer pairs amplified a hypervariable region of the mitochondrial 12S rRNA gene (ca. 172 bp; hereafter called the 'MiFish sequence') and appended primer-binding sites (5′ ends of the sequences before six random bases [Ns]) for sequencing at both ends of the amplicon. Six Ns in the middle of these primers enhanced cluster separation on the flow cells during initial base-call calibrations on the MiSeq platform.

The 1st PCR consisted of 35 cycles with a 12-µl reaction volume containing 6.0 µl of 2 × KAPA HiFi HotStart ReadyMix (KAPA Biosystems, MA, USA), 2.8 µl of a mixture of the three MiFish primer sets at a volume ratio of 2:1:1 (U:E:U2 forward and reverse primers; 5 µM), 1.2 µl of sterile distilled $H_2O$, and 2.0 µl of the eDNA template. To minimize PCR dropouts during the 1st PCR (*Doi et al., 2019*; *Miya et al., 2020a*), we performed eight technical replicates for the same eDNA template using a strip of eight tubes (200 µl). The thermal cycle profile after an initial 3 min denaturation at 95°C was as follows: denaturation at 98°C for 20 s, annealing at 65°C for 15 s, and extension at 72°C for 15 s, with a final extension at the same temperature for 5 min. We also prepared a 1st PCR blank (1B) during this process, in addition to FB and EB. However, we did not perform eight replications and only used a single tube for each of the three blanks (FB, EB, and 1B) to minimize costs.

After completing the 1st PCR, we pooled an equal volume of PCR products from each of the eight replicates in a single 1.5-ml tube. We purified pooled products using a GeneRead Size Selection kit (QIAGEN, Hilden, Germany) following the manufacturer's GeneRead DNA Library Prep I Kit protocol. The column purification process was repeated twice. We subsequently quantified the purified target products (ca. 300 bp) using TapeStation 2200 (Agilent Technologies, Tokyo, Japan), diluted to 0.1 ng/µl using Milli Q water, and used the diluted products as templates for the second-round PCR (2nd PCR). Regarding the three blanks (FB, EB, and 1B), we purified the 1st PCR products in the same manner, but did not quantify the purified PCR products. We instead diluted them according to an average dilution ratio for positive samples, following which we used the diluted products as templates for the 2nd PCR.

In the 2nd PCR, we used the two primers to append dual-indexed sequences (eight nucleotides indicated by **X**) and flow cell-binding sites for the MiSeq platform: 2nd PCR-forward and 2nd PCR-reverse (*Table 1*). We performed the 2nd PCR with 10 cycles of a 15-µl reaction volume containing 7.5 µl of 2 × KAPA HiFi HotStart ReadyMix, 0.9 µl of each primer (5 µM), 3.9 µl of distilled $H_2O$, and 1.9 µl of the template (0.1 ng/µl with the exceptions of the three blanks). The thermal cycle profile after an initial 3 min denaturation at 95°C was as follows: denaturation at 98°C for 20 s, annealing and extension combined at 72°C (shuttle PCR) for 15 s with the final extension at the same temperature for 5 min. We also made a 2nd PCR blank (2B) during this process in addition to FB, EB, and 1B. In total, we made 195 blanks (FB = 60, EB = 50, 1B = 50, 2B = 35) and subjected them to the above library preparation procedure to monitor contamination during the on-site filtration, subsequent DNA extraction, and 1st and 2nd PCR of the 550 samples.

We adjusted the number of samples per MiSeq sequencing to obtain approximately 100,000 reads per sample. We pooled each of the individual paired-end libraries in an equal volume into a 1.5-ml tube. We then electrophoresed the pooled libraries using a 2% E-Gel Size Select agarose gel (Invitrogen, CA, USA) and excised the target amplicons (ca. 370 bp). The concentrations of the size-selected libraries were measured using a Qubit dsDNA HS assay kit and Qubit fluorometer (Life

Technologies, CA, USA), diluted to 10–12.0 pM with HT1 buffer (Illumina, CA, USA), and sequenced on the MiSeq platform using a MiSeq v2 Reagent Kit for 2 × 150 bp PE (Illumina, CA, USA) with a PhiX Control library (v3) spike-in (expected at 5%) following the manufacturer's protocol. We performed 21 MiSeq runs (with eDNA samples of other projects) to complete the analysis, which generated 71,892,685 reads in total for the eDNA samples of this project.

## Sequence analysis

We performed data preprocessing and analyses of raw MiSeq reads from the MiSeq run using PMiFish ver. 2.4 (*Miya et al., 2020b*; *Miya et al., 2020a*) according to the following steps: (1) Forward (R1) and reverse (R2) reads were merged by aligning the two reads using the *fastq merge pairs* command. During this process, short reads (<100 bp) after tail trimming and paired reads with too many differences (>5 positions) in the aligned region (ca. 65 bp) were discarded; (2) primer sequences were removed from merged reads using the *fastx truncate* command; (3) reads without primer sequences underwent quality filtering using the *fastq filter* command to remove low-quality reads with an expected error rate of >1% and short reads of <20 bp; (4) preprocessed reads were dereplicated using the *fastx uniques* command and all singletons, doubletons, and tripletons were removed from subsequent analyses to avoid false positives following the recommendation by the author of the program (*Edgar, 2010*); (5) dereplicated reads without singletons, doubletons, and tripletons were denoised using the *unoise3* command to generate amplicon sequence variants (ASVs) (*Callahan et al., 2017*; *Edgar, 2016*); (6) ASVs were rarefied to the approximate minimum read number (20,000), which resulted in 10,984,789 reads in total for the rarefied ASV table; and (7) ASVs were subjected to taxon assignments to species names (molecular operational taxonomic units; MOTUs) using the *usearch global* command with a sequence identity of >98.5% with the reference sequences (two nucleotide differences allowed) and a query coverage of ≥90%.

## Taxon assignment

ASVs with sequence identities of 80–98.5% were tentatively assigned 'U98.5' labels before the corresponding species names with the highest identities (e.g., U98.5 *Pagrus major*) and were subjected to clustering at the level of 0.985 using the *cluster smallmem* command. An incomplete reference database necessitates this clustering step, which enables the detection of multiple MOTUs for identical species names. Multiple MOTUs were annotated as 'gotu1, 2, 3…' and all outputs (MOTUs plus U98.5 MOTUs) were tabulated with read abundance. ASVs with sequence identities of <80% (saved as 'no hit') were excluded from the above taxon assignments and downstream analyses because they were all non-fish organisms. MiFish DB ver. 43 was used for taxon assignment, comprising 7973 species distributed across 464 families and 2675 genera. At present, the reference sequences are available for about 70% of 4500 fish species in Japan. However, due to the unknown degree of intraspecific variation, using a uniform threshold of 98.5% to delineate species can result in over-splitting or over-clustering MOTUs. To solve this issue, manual refinement of the taxon assignments was performed based on the phylogenetic tree.

To refine the above taxon assignments, family-level phylogenies were reproduced from MiFish sequences from MOTUs, U98.5 MOTUs, and reference sequences (contained in the MiFish DB ver. 43) belonging to these families. In each family, representative sequences (most abundant reads) from MOTUs and U98.5 MOTUs were assembled, and all reference sequences were added from that family and saved in the FASTA format. Combined FASTA-formatted sequences were subjected to multiple alignments using MAFFT 7 (*Katoh and Standley, 2013*) with a default set of parameters. A neighbor-joining (NJ) tree was subsequently constructed with the aligned sequences in MEGA X (*Stecher et al., 2020*) using Kimura two-parameter distances. Distances were calculated using the pairwise deletion of gaps and among-site rate variations modeled with gamma distributions (shape parameter = 1). Furthermore, bootstrap resampling (*n* = 100) was performed to estimate statistical support for the internal branches of the NJ tree and midpoint rooting was conducted on the resulting NJ tree.

A total of 103 family-level trees were visually inspected and taxon assignments were revised in the following manner. Among U98.5 MOTUs placed within a monophyletic group consisting of a single genus, unidentified MOTUs were named after that genus, followed by 'sp.' with sequential numbers (e.g., *Pagrus* sp. 1, sp. 2, sp. 3...). Regarding the remaining MOTUs ambiguously placed in the family-level tree, unidentified MOTUs were named after that family, followed by 'sp.' with

sequential numbers (e.g., Sparidae sp. 1, sp. 2, sp. 3...). The final list of detected species is provided in *Supplementary file 1a*. The negative controls produced negligible reads and all of the reads were assigned to non-target taxa. Therefore, we discarded the sequence reads from the negative control samples (see Results and Discussion for details).

## qPCR and estimation of DNA copy numbers

The nonlinear time series analytical tools used in the present study require a quantitative time series, namely, relative abundance data that are common for eDNA metabarcoding studies are not suitable. To estimate fish eDNA concentrations, we initially quantified the eDNA concentrations of the most common fish species in the region, Japanese black seabream (*A. schlegelii*), using qPCR. Metabarcoding detected the eDNA of *A. schlegelii* in 504 out of 550 water samples (91.6%) (*Figure 1—figure supplement 1*), and, thus, we decided to use the sequence reads of *A. schlegelii* as the internal standard DNA of each sample.

We performed qPCR using the LightCycler 96 System (Roche Diagnostics, Mannheim, Germany). Twenty microliters of the PCR reaction mixture comprised 2 μl of the template DNA extract, a final concentration of 900 nM each of the forward (5′-CTG TCT GCC GTC CCC TAC A-3′) and reverse (5′-TAT GGC GGC TAC GAT AAA AGG A-3′) primers, a final concentration of 125 nM of the probe (5′-FAM-TCA GTT GAC AAC GCA ACC CTA ACC CG-TAMRA-3′), and 1 × PCR master mix (FastStart Essential DNA Probes Master, Roche). The primers and probe were designed to specifically amplify a 129-bp fragment from the *A. schlegelii* mitochondrial cytochrome *b* (cyt *b*) gene (*Takahashi et al., 2020*). The species specificity of the primers and probe was checked by *Sasano et al., 2022*. Reaction conditions were as follows: 10 min at 95°C, 55 cycles at 95°C for 10 s, and at 60°C for 30 s. PCR triplicates were provided for each sample. We quantified the concentrations of DNA from the calibration curve obtained from amplifying triplicates of the four standards containing $3 \times 10$, $3 \times 10^2$, $3 \times 10^3$, and $3 \times 10^4$ copies of artificial DNA fragments inserted into the target region. To check cross-contamination during the PCR process, we also used triplicates of a mixture that contained 2 μl of pure water instead of template DNA. We did not detect *A. schlegelii* DNA from any FB, EB, or PCR blanks by qPCR.

We converted eDNA sequence reads obtained by eDNA metabarcoding using the quantities of *A. schlegelii* eDNA as internal standard DNAs. By dividing *A. schlegelii* sequence reads by the *A. schlegelii* eDNA quantity in each sample, we estimated the number of sequence reads generated per *A. schlegelii* eDNA copy for each sample, and we applied the same conversion factor (i.e., sequence reads/*A. schlegelii* eDNA copy) to other fish species. Note that sequence reads generated per eDNA copy may vary depending on factors such as the level of PCR inhibition and PCR amplification efficiency; however, in general, this 'internal spike-in DNA' method has been shown to reasonably estimate the quantity of eDNA concentrations (*Ushio et al., 2018b*; *Ushio, 2022a*). Species-specific biases that may be introduced in the internal spike-in DNA method do not cause serious biases in the outcomes of our nonlinear time series analysis because it standardizes time series to have a zero mean and a unit variance before analyses and also only utilizes the fluctuation patterns of time series. When we did not detect any *A. schlegelii* eDNA by qPCR, we replaced the 'zero' value with the minimum eDNA copy numbers of *A. schlegelii* (0.346 copies/μl DNA). Similarly, when we did not detect any *A. schlegelii* eDNA sequence reads by metabarcoding, we replaced the 'zero' value with the minimum eDNA sequence reads of *A. schlegelii* (12 reads/sample). These corrections estimated how many sequence reads were generated per eDNA copy for all samples.

## Nonlinear time series analysis to detect fish–fish interactions

We detected fish–fish interactions using a nonlinear time series analysis based on quantitative eDNA time series data from multiple species and sites. Since the reliable detection of interspecific interactions requires sufficient information in the target time series, we selected the top 50 most frequently detected species and excluded other rarer fish species from the analysis. We quantified information flow between two fish species' eDNA time series by the 'unified information-theoretic causality (UIC)' method (*Osada et al., 2023*) implemented in the 'rUIC' v0.1.5 package (*Osada and Ushio, 2021*) of R (*R Development Core Team, 2022*). UIC tests the statistical clarity of information flow between variables in terms of TE (*Schreiber, 2000*) computed by nearest neighbor regression based on time-delay embedding of explanatory variables (i.e., cross mapping; *Sugihara et al., 2012*) (as for the term

'statistical clarity', see the section below). In contrast to the standard method used to measure TE, UIC quantifies information flow as follows:

$$TE = \frac{1}{T} \sum_{t=1}^{T} \log \left( \frac{p\left(y_{t+tp} \mid x_t, x_{t-\tau}, \cdots, x_{t-(E-1)\tau}, z_t\right)}{p\left(y_{t+tp} \mid x_{t-\tau}, x_{t-2\tau}, \cdots, x_{t-(E-1)\tau}, z_t\right)} \right), \qquad (1)$$

where $x$, $y$, and $z$ represent an effect variable, a potential causal variable, and a conditional variable (if available), respectively. $p\left(A \mid B\right)$ represents conditional probability: the probability of $A$ conditioned on $B$. $t$, $tp$, $\tau$, and $E$ represent the time index, time step, a unit of time-lag, and the optimal embedding dimension, respectively. $T$ is the total number of points in the reconstructed state space (this is equivalent to the total number of time points – the optimal embedding dimension + 1). For example, if $tp$ = −1 in **Equation 1**, UIC tests the causal effect from $y_{t-1}$ to $x_t$. Optimal $E$ was selected by measuring TE as follows:

$$TE = \frac{1}{T} \sum_{t=1}^{T} \log \left( \frac{p\left(x_{t+tp} \mid y_t, x_t, x_{t-\tau}, \cdots, x_{t-(E-1)\tau}\right)}{p\left(x_{t+tp} \mid y_t, x_t, x_{t-\tau}, \cdots, x_{t-(E_R-1)\tau}\right)} \right), \qquad (2)$$

where $E_R$ $(< E)$ is the optimal embedding dimension of lower dimensional models. **Equation 2** is a TE version of simplex projection (**Sugihara and May, 1990**), and if $tp$ = 1, it determines the optical $E$ based on one-step forward prediction. In the present study, the causality time-lag ($tp$ in **Equation 1**) up to −6 (equivalent to 3-month time-lag) was tested. Statistical clarities were tested by bootstrapping data after embedding (the threshold was set to 0.05). TE measured according to **Equation 1** gains the advantage of previous causality tests, that is, standard TE methods (**Runge et al., 2012**; **Schreiber, 2000**) and CCM (**Sugihara et al., 2012**), the algorithm of which is explained in **Osada et al., 2023** and implemented in **Osada and Ushio, 2021**.

By using UIC, we quantified TE between fish eDNA time series. As environmental variables, water temperature (°C), salinity (‰), wave height (m), and tide level (cm) were considered as conditional variables. If the environmental variables had statistically clear influence on fish eDNA dynamics, they were included in the calculation of TE as $z_t$ in **Equation 1**. This means that the effects of the environmental variables on the fish eDNA abundance were removed in the analysis when detecting interspecific interactions. Importantly, in most cases, water temperature had statistically clear influence on fish eDNA dynamics and included as a conditional variable ($z_t$) in the embedding. In addition, although water temperature showed clear seasonality in the region (**Figure 1—figure supplement 2**), including water temperature as a conditional variable ($z_t$) took the effect of the seasonality in detecting causation into account. We merged all eDNA time series across the 11 study sites and standardized the eDNA time series to have zero means and a unit of variance before the analysis. If TE between two fish eDNA time series was statistically clearly higher than zero, we interpreted it as a sign of an interspecific interaction. After the detections of interspecific interactions among the top 50 fish species, we visualized the interaction network (i.e., the reconstruction of fish–fish interaction network). Importantly, UIC quantifies the average information flow between two time series (**Equation 1**), and thus there is only one TE value for each pair of fish species, which is critically different from interaction strengths quantified by the MDR S-map (see the following section).

## Nonlinear time series analysis to quantify interaction strengths

We quantified fish–fish interaction strengths for the interspecific interactions detected by UIC. For this purpose, we used an improved version of the sequential locally weighted global linear map (S-map) (**Sugihara, 1994**), called the multiview distance regularized S-map (MDR S-map) (**Chang et al., 2021**). Consider a system that has $E$ different interacting variables (time-delay coordinates may be included), and assume that the state space at time $t$ is given by $\boldsymbol{x}_t = \{x_{1,t}, x_{2,t}, ..., x_{E,t}\}$. For each target time point $t^*$, the S-map method produces a local linear model that predicts the future value $x_{1,t^*+tp}$ from the multivariate reconstructed state space vector $\boldsymbol{x}_{t^*}$. That is,

$$\hat{x}_{1,t^*+tp} = IS_0 + \sum_{j=1}^{E} IS_j \, x_{j,t^*} \qquad (3)$$

where $\hat{x}_{1,t^*+tp}$ is a predicted value of $x_1$ at time $t^* + tp$, $IS_j$ is a regression coefficient and interpreted as interaction strength (or called S-map coefficient), and $IS_0$ is an intercept of the linear model. The linear model is fit to the other vectors in the state space. However, points that are close to the target point, $\boldsymbol{x}_{t^*}$, are given greater weighting (i.e., locally weighted linear regression). In the standard S-map, the distances between the target point and other points are measured by Euclidean distance. However, Euclidean distance cannot be a good measure if the dimension of the state space is high, and in this case, it is impossible to identify nearest neighbors correctly. In the MDR S-map, the distance between the target point and other points is measured by the multiview distance (*Chang et al., 2021*), which can be calculated by ensembling distances measured in various low-dimensional state spaces (multi-view embedding; see *Ye and Sugihara, 2016*). In addition, to reduce the possibility of overestimation and to improve forecasting skill, regularization (i.e., ridge regression) is also applied (*Cenci et al., 2019*). *Chang et al., 2021* showed that the MDR S-map outperformed other S-map methods and it enables improved estimations of interaction strengths. Importantly, the MDR S-map quantifies interaction strength at each time point (i.e., the maximum possible number of interaction strength values for each fish–fish interaction is the number of time points – the optimal embedding dimension + 1). As in the UIC analysis, we included temperature or other environmental variables as conditional variables if they had statistically clear influence on eDNA dynamics of a particular fish species. In the present study, we implemented the MDR S-map in our custom R package, 'macam' v0.1.3 (*Ushio, 2022b*) and used it for computation.

## Quantification of the temperature sensitivity of fish species interactions

After reconstructing fish interaction networks and quantifying interaction strengths, we analyzed the relationships among fish species interaction strengths and biotic and abiotic variables. We performed all analyses using R v4.2.1 (*R Development Core Team, 2022*). Generalized additive mixed models (GAMMs) were performed using the 'mgcv' package of R (*Wood, 2004*). We visualized results with 'ggplot2' (*Wickham, 2009*) and 'cowplot' (*Wilke, 2017*). In the present study, we used the term 'statistical clarity' instead of 'statistical significance' to avoid misinterpretations, according to the recommendations by *Dushoff et al., 2019*.

In the first analysis, the relationships between fish–fish interaction strengths and the characteristics of the study sites were analyzed by GAMM. In the analysis, in-strength (interaction strengths that a species receives) and out-strength (interaction strengths that a species gives) were separately calculated. In GAMM, in-strength or out-strength were an explained variable, and an environmental or ecological variable was an explaining variable. Explaining variables included water temperature, species richness, total fish eDNA concentrations, salinity, tide level, and wave height. A gamma distribution was assumed as the error distribution, the 'log' function was used as a link function, and the study site and fish species were used as a random effect (i.e., in R, gamm(abs(IS) ~ s(explaining_variable), family = Gamma(link = 'log'), random = list(site = ~1, fish_species = ~1)), where s() indicates a smoothing term). The biological assumption behind this modeling is that explaining variables, such as water temperature, may linearly or nonlinearly influence fish species interactions. Moreover, the effects of explaining variables may randomly vary depending on the study site and fish species. The effect was considered to be statistically clear at $p < 0.05$.

In the second analysis, we focused on the effects of water temperature on the interaction strengths of each fish. In the analysis, GAMM was used again. A gamma distribution was assumed as the error distribution and the 'log' function was used as a link function, and the study site was used as a random effect (i.e., in R, gamm(abs(IS) ~ s(explaining_variable), family = Gamma(link="log"), random = list(site = ~1)), where s() indicates a smoothing term). GAMM was separately performed for each fish species. We also analyzed the effects of species richness and total eDNA concentration on the interaction strengths of each fish species. Again, the effect was considered to be statistically clear at $p < 0.05$.

## Results and discussion
### Taxonomic diversity and dynamics of fish eDNA
Our multiple MiSeq runs generated 71,892,685 reads in total for the eDNA samples, of which 98% were assigned to fish species, and detected 1,130 MOTUs. We inspected their family-level phylogenies

to increase the accuracy of taxonomic assignments, recognizing 856 MOTUs across 33 orders, 167 families, and 466 genera (*Supplementary file 1a*). This taxonomic diversity was similar to that of the local fauna (948 species across 33 orders, 158 families, and 493 genera) compiled from a literature survey and museum collections (*Supplementary file 1b*). The negative controls produced negligible reads (177 ± 665 reads [mean ± standard deviation]), which accounted for ca. 0.1% of the positive sample reads. Moreover, all of the reads were assigned to non-target taxa, such as fish species that had never been observed in the study region and freshwater fish species (possibly contaminated from the laboratory). Therefore, we conclude that any contaminations in our experiments were negligible, and we discarded the sequence reads from the negative control samples.

Sequence reads in the rarefied ASV table were converted to estimated eDNA concentrations by using the eDNA concentrations of a common fish species detected across most samples (Japanese Black Seabream, *A. schlegelii*) (i.e., an analog of the internal spike-in DNA method; see Methods and *Figure 1—figure supplement 1*). The converted ASV table included 23,863 detections and we selected the top 50 most frequently detected species for our time series analysis. The detection frequencies of these 50 species ranged between 148 and 532 with a mean of 296, and their total detection frequencies reached 14,793 (62.0% of the total detection frequency).

Fish eDNA concentrations and fish species richness showed a clear intra-annual pattern (i.e., seasonality; *Figure 1b, c*), which were higher in warmer months (e.g., between July and October)

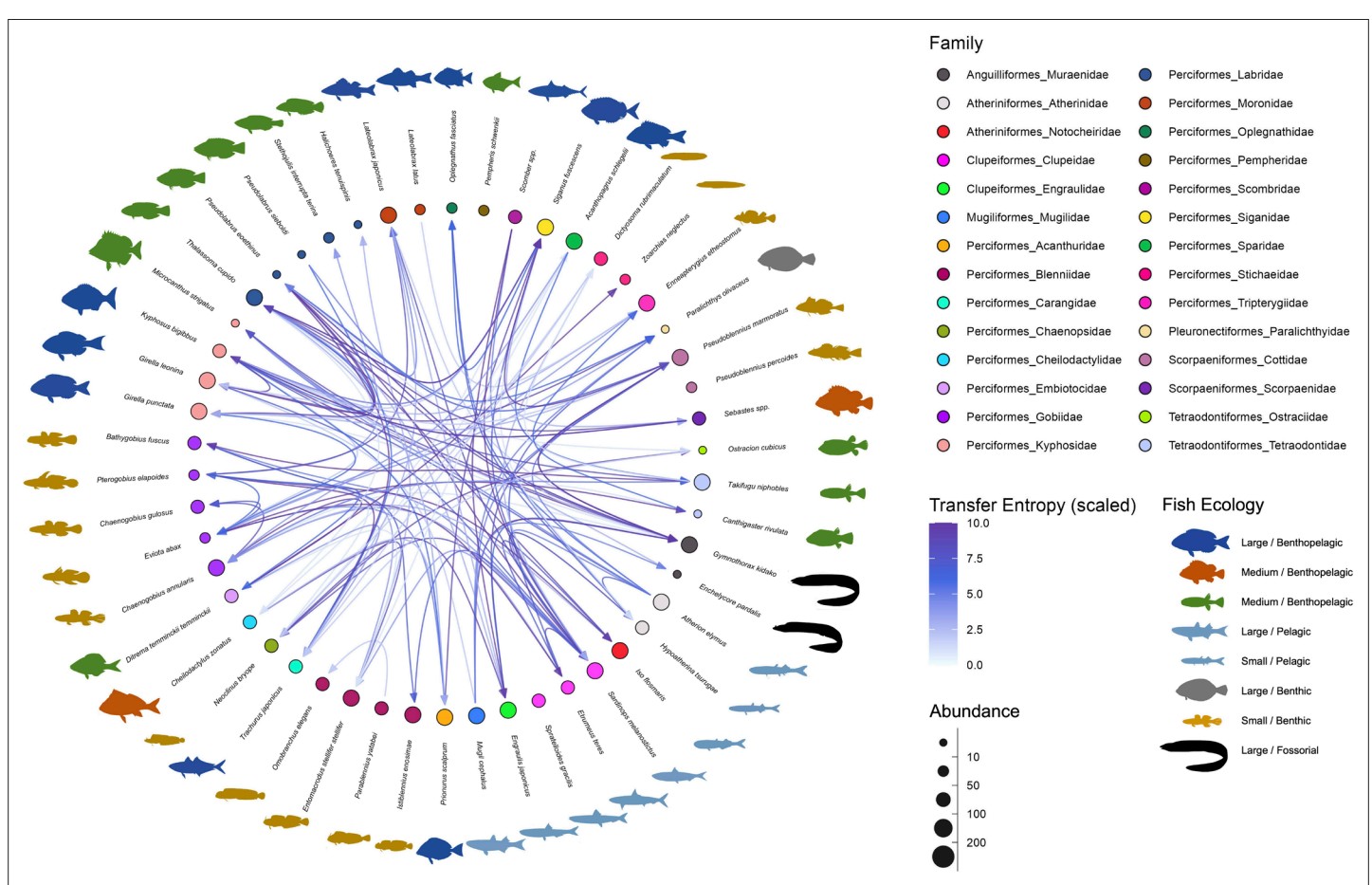

**Figure 2.** Interaction networks of the fish community in the Boso Peninsula coastal region. The 'average' interaction network reconstructed by quantifying information transfer between environmental DNA (eDNA) time series. Transfer entropy (TE) was quantified by leveraging all eDNA time series from multiple study sites to draw this network. Only information flow larger than 80% quantiles (i.e., strong interaction) was shown as interspecific interactions for visualization. The edge color indicates scaled TE values, and fish illustration colors represent their ecology (e.g., habitat and feeding behavior). Node colors and node sizes indicate the fish family and fish abundance (total eDNA copy numbers of the fish species), respectively.

The online version of this article includes the following figure supplement(s) for figure 2:

**Figure supplement 1.** The relationships between network properties and environmental variables.

than in colder months (e.g., between January and March) at all sites. Total eDNA concentrations and species richness positively correlated with water temperature (*Figure 1—figure supplement 2*). Seasonal changes in eDNA concentrations were consistent with the patterns of seasonal occurrences in tropical and subtropical fish species in the Boso Peninsula, to which they are transported by the warm Kuroshio Current, settling on the coastal waters during the warmer months and subsequently disappearing during the colder months (*Saito, 2019*; *Senou et al., 2006*).

## Reconstruction of the fish species interaction network

We reconstructed fish species interaction networks based on the quantitative, multispecies fish eDNA time series. Regarding the 50 frequently detected fish species, we quantified pairwise information flow between fish species. *Figure 2* shows reconstructed fish interaction networks for the 50 frequently detected fish species in the Boso Peninsula. At the regional scale, the linear correlations among water temperature, total eDNA concentrations, interaction strengths, the number of interactions, and species richness was all statistically clear (p < 0.05; *Figure 2—figure supplement 1*) except for the linear correlation between water temperature and mean interaction strengths (p > 0.05). The interaction strengths became weak as species richness increased, which is consistent with a previous study (*Ratzke et al., 2020*; *Ushio, 2022a*), suggesting that understanding the causes and effects of weak interactions is key to understanding the maintenance of species-rich communities.

Most of the statistically clear information flow may be interpreted from the viewpoint of fish–fish behavior-level interspecific interactions (*Supplementary file 1c*), which is convincing considering the time resolution of our eDNA time series (but see Potential limitations of the present study). The largest information flow was detected from *Pseudoblennius marmoratus* to *Pseudolabrus eoethinus* (*Supplementary file 1c*). These fish species have overlapping habitats, and thus, they may interact. Bidirectional information flow was detected between *P. eoethinus* and *Gymnothorax kidako*. They are both carnivorous fish species and may conduct joint hunting. *Chaenogobius annularis* and *Takifugu niphobles* are often found together in the sand between reefs. *T. niphobles* frequently dive in the sand, and benthos and mysids that are dug up during the dive may be prey for *C. annularis*. Further interpretations about the detected information flow are described in *Supplementary file 1c*.

## Interaction strengths and environmental variables

We investigated how interaction strengths (i.e., regression coefficients estimated by the MDR S-map) changed with environmental variables (e.g., water temperature) and ecological properties (e.g., species richness and total eDNA concentration) at the community level (*Figure 3* and *Figure 3—figure supplement 1*). The in-strengths and out-strengths of fish species interactions were statistically clearly associated with water temperature, species richness, and total eDNA concentration (GAMM, p < 0.05; *Figure 3* and *Supplementary file 1d*) except for the effects on species richness on the in-strengths (*Figure 3b*). In-strengths of the fish–fish interactions increased with water temperature (*Figure 3a*), supporting our first hypothesis while out-strengths of the interactions showed an opposite pattern (*Figure 3d*), which might suggest there is a difference in the temperature dependence between the in-strengths and out-strengths of the interactions. Indeed, water temperature may influence fish physiological activity (*Claireaux et al., 2006*; *Kishi et al., 2005*; *Oyugi et al., 2012*) often in a complex way, and thus, the community-level influence of water temperature may also be complex as they should arise from the individual-level influence of water temperature on fish. Interaction strengths were also statistically clearly influenced by species richness and total eDNA concentrations (except for *Figure 3b*); the interaction strengths decreased with increasing species richness and total eDNA concentration. The effects of salinity, tide and wave were less clear, although the effects were statistically clear except for the effects of tide level on the out-strength (*Figure 3—figure supplement 1*). Overall, environmental and ecological variables influenced the interaction strengths statistically clearly, but large variations remained unexplained (*Figure 3—figure supplements 2 and 3*), suggesting that other factors (e.g., fish ecology, nutrient, and physiological status) may influence the interaction strengths.

We also investigated how temperature influenced the interaction strength of individual fish species. *Figure 4* shows how interaction strengths among fish species changed with water temperature. For visualization purpose, we show fish species of which interaction strengths changed with water temperature highly statistically clearly (p < 0.0001). The in-strengths of several fish species clearly

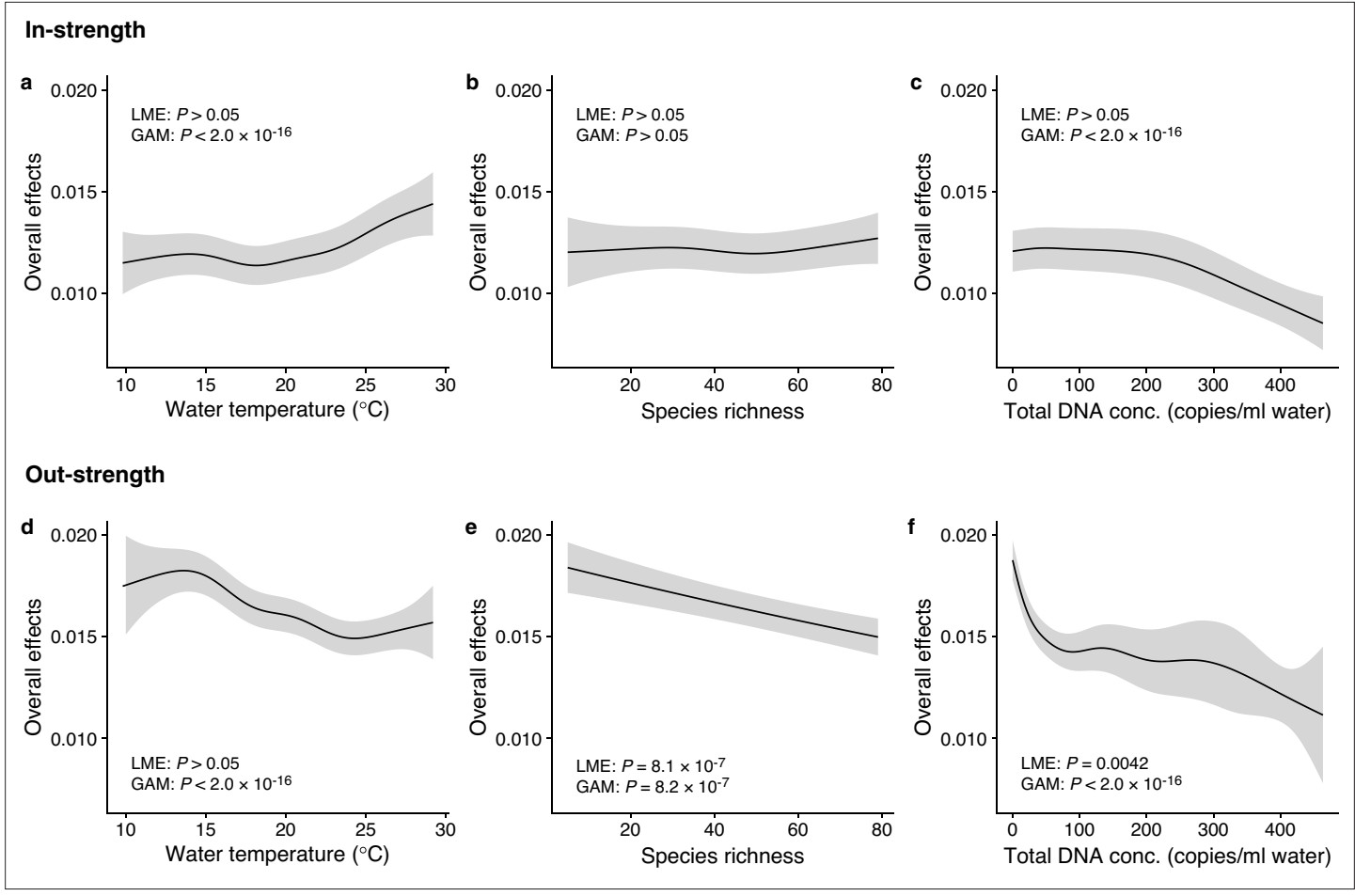

**Figure 3.** Dependence of interaction strengths on biotic and abiotic variables (50 dominant fish species and 11 study sites were leveraged). The panels show the overall effects of biotic and abiotic variables on interaction strengths of the 50 dominant fish species: Effects of (**a, d**) water temperature, (**b, e**) species richness, and (**c, f**) total environmental DNA (eDNA) copy numbers. The *y*-axis indicates the effects of the variables on fish–fish interaction strengths quantified by the MDR S-map method. (**a–c**) show the effects on the species interactions that a focal species receives (i.e., in-strength), and (**d–f**) show the effects on the species interactions that a focal species gives (i.e., out-strength). The line indicates the average effects estimated by the general additive model (GAM), and the gray region indicates 95% confidential intervals. LME and GAM indicate the statistical clarity of the linear mixed model portion and GAM portion, respectively. Detailed statistical results and raw data are shown in **Supplementary file 1d** and **Figure 3—figure supplement 1**, respectively.

The online version of this article includes the following figure supplement(s) for figure 3:

**Figure supplement 1.** Dependence of interaction strengths on additional abiotic variables.

**Figure supplement 2.** The relationship between interaction strengths and water temperature, species richness, and total DNA concentrations.

**Figure supplement 3.** The relationship between interaction strengths and salinity, tide level, and wave.

increased at higher water temperatures: for example, *Halichoeres tenuispinis*, *Macrocanthus strigatus*, *Pempheris schwenkii*, *Stethojulis interrupta terina*, and *Thalassoma cupido* (**Figure 4a**). On the other hand, in-strengths of some fish species and out-strengths decreased at higher water temperatures: for example, in-strengths of *Ditrema temminckii temminckii*, *Engraulis japonicus*, and *Girella punctata*, and out-strengths of five fish species (**Figure 4a, b**). These results support our second hypothesis that the relationship between the patterns of temperature and interaction strengths varies among fish species. These results suggest that, although water temperature may have strong influences on fish–fish interaction strengths in general, the sign of temperature effect (i.e., positive or negative) can vary depending on fish species and environmental conditions. Previous studies showed that fish physiological activities, such as feeding rates, growth rates, and swimming speed, were influenced by water temperature (*Claireaux et al., 2006*; *Kishi et al., 2005*; *Oyugi et al., 2012*). In addition, the direction (i.e., positive or negative) of the temperature effects on fish metabolic activities may be

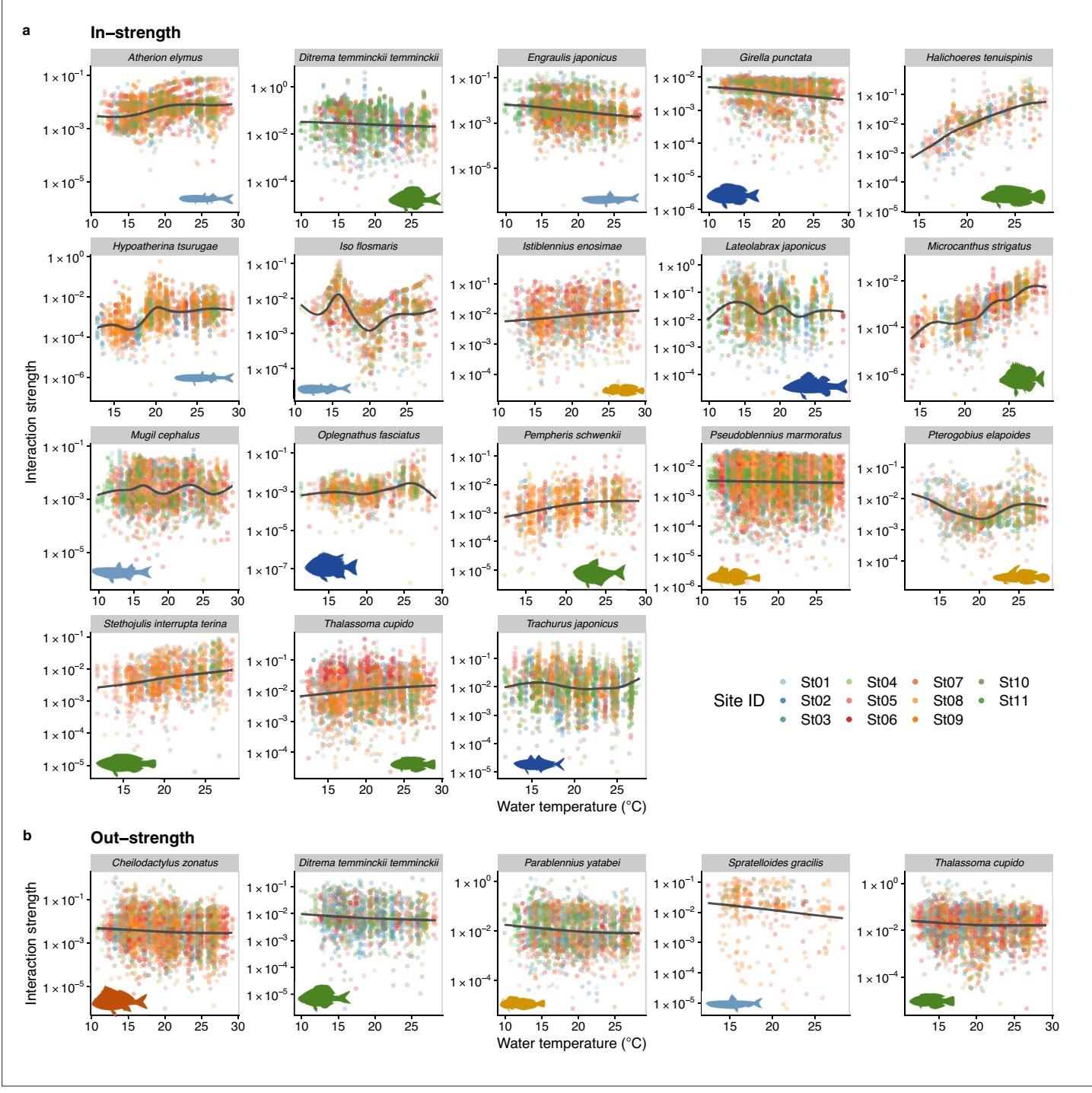

**Figure 4.** Temperature dependence of fish species interactions at the species level. (**a** and **b**) show temperature effects on fish species interactions quantified by the MDR S-map method. Note that the MDR S-map enables quantifications of interaction strengths at each time point, and thus the number of data points is large. (**a**) Points indicate the species interactions that a focal species (indicated by the strip label and fish image) receives (i.e., in-strength). (**b**) Points indicate the species interactions that a focal species (indicated by the strip label and fish image) gives (i.e., out-strength). For (**a**) and (**b**), only fish species of which interactions are statistically clearly affected by water temperature are shown (to exclude fish species with relatively weak temperature effects, $p < 0.0001$ was used as a criterion here). Point color indicates the study site. Gray line is drawn by general additive model (GAM; the study sites were averaged for visualization purpose).

The online version of this article includes the following figure supplement(s) for figure 4:

**Figure supplement 1.** Dependence of fish species interactions on species richness at the fish species level.

**Figure supplement 2.** Dependence of fish species interactions on the total DNA concentration (an index of total fish abundance) at the species level.

species specific (*Oyugi et al., 2012*), and this species specificity may underlie the species-specific effects of temperature on the interaction strength.

## Potential limitations of the present study and future perspectives

The present results showing that the strengths of fish–fish interactions depend on water temperature rely on several assumptions that were not fully investigated, and thus, careful interpretations and discussions are required. First, the extent to which fish eDNA concentrations accurately represent fish abundance is an open question. Previous studies demonstrated that the quantity of eDNA may be a proxy of fish abundance and/or biomass (i.e., a positive correlation between the quantity of eDNA and fish abundance/biomass) (*Takahara et al., 2012*; *Thomsen et al., 2012*; *Yamamoto et al., 2016*). Our nonlinear time series analysis used the information embedded in fluctuations in time series, and time series are always standardized before the analysis. Therefore, species-specific differences in the copy numbers of the genetic marker and eDNA release rates, which may bias the estimation of absolute abundance, do not cause serious biases in the reconstruction of the networks and estimations of interaction strengths. However, accurate estimations of the absolute abundance of fish will improve the accuracy of analyses, and the integration of eDNA concentration data, knowledge on eDNA dynamics (i.e., release and degradation rates), and the hydrodynamic modeling of ocean water flow will be a promising approach (e.g., as in *Fukaya et al., 2021*).

Another limitation is that the reconstructed interaction network was not fully validated in the present study. Although previous studies demonstrated that network reconstructions based on a time series analysis work reasonably well and show meaningful patterns (*Runge et al., 2019*; *Sugihara et al., 2012*; *Ushio, 2022a*; *Ushio et al., 2018a*), experiment/observation-based validations of the reconstructed network have not yet been performed. Potential causal interactions between fish species may be reasonably interpreted (*Supplementary file 1c*); however, developing a framework to efficiently validate 'previously unknown' interactions detected by eDNA data and a nonlinear time series analysis is an important future direction.

Third, although we showed that temperature effects on interaction strength are statistically clear and common in the Boso Peninsula coastal region, the range of water temperature in the present analysis was still not broad and there is currently no evidence that we can generalize these results to other regions. Nonetheless, the advantages of the eDNA method are the low cost and minimal time and labor needed for field sampling as well as the potential scalability of the library preparation process (*Ushio et al., 2022c*). Therefore, frequent (*Ushio, 2022a*) and large spatial-scale ecological monitoring (e.g., ANEMONE DB; https://db.anemone.bio/) is possible, which will provide a more detailed understanding of the temperature interaction strength relationships of fish on a larger spatial scale.

Lastly, other environmental conditions may covary with temperature and influence fish–fish interactions in a species-specific way (*Figure 4—figure supplements 1 and 2*). Although the effect of temperature on the interaction strengths was one of the strongest at the community level among the environmental variables (*Supplementary file 1d*), there is a possibility that the temperature effect we observed might not be direct. For example, water temperature and oxygen concentration covary and both directly/indirectly affect fish physiology (*Salvatteci et al., 2022*). Because direct field manipulations to validate our results are challenging, robust conclusions about the temperature sensitivity of fish interactions may only be made by integrating results from different approaches (e.g., small-scale lab experiments and time series-based causal analysis). In nature, multiple environmental variables are continuously changing, and the interaction strengths may fluctuate through time and space being affected by the environmental variables, as shown in previous studies (*Ushio, 2022a*; *Ushio et al., 2018a*). Thus, interactions detected and quantified under a controlled environment might not necessarily be observed under field conditions. Our research framework that enables the detection and quantification of interactions in nature provides a complementary view about fish–fish interactions, which would play a critical role in understanding the effects of temperature on fish–fish interactions.

## Implications for fish community assembly and the effect of global climate change

Water temperature has significant influences on marine community composition and diversity at the global spatial scale and historical time scale (*Tittensor et al., 2010*; *Yasuhara and Deutsch, 2023*), but the mechanism of temperature effects on fish community assembly is not fully understood. Recent

studies have shown that temperature (and oxygen) plays an important role in determining fish body size at the historical time scale (*Salvatteci et al., 2022*; *Yasuhara and Deutsch, 2022*). As fish body size plays a critical role in interspecific interactions such as predator–prey interactions (e.g., predator–prey mass ratio; *Nakazawa et al., 2011*), temperature-induced body size changes may influence interspecific interactions at a longer time scale. On the other hand, our study showed that temperature may induce changes in fish–fish interactions at a relatively short time scale (weeks to months), perhaps via temperature effects on the physiological activity of fish individuals. These suggest that temperature effects on fish community assembly involve effects at different time scales, and thus, integrating results from different temporal (and spatial) scales are necessary to understand fish community assembly processes in nature.

In addition, our study revealed that temperature effects on fish–fish interactions depend on fish species identity. This suggests that, even in the same habitat, the temperature sensitivity of fish–fish interactions is variable and fish species specific, and that consequences of changing temperature in the community assembly process may be complex. For example, increased water temperature strengthens interactions received by *H. tenuispinis* and *M. strigatus* (*Figure 4*), which may destabilize the population dynamics of these species. In contract, increased water temperature may exert the opposite effects on *E. japonicus* and *G. punctata* (*Figure 4*), that is, weakened interactions and stabilized population dynamics. How these varying responses to temperature change, or 'response diversity' (*Ross et al., 2023*), influences overall community dynamics remains unclear. Our study provides a practical framework to quantify response diversity using time series (the S-map method calculates the first derivative as an interaction strength, and it is an analog of the additive model-based method proposed by *Ross et al., 2023*), and quantifying species-specific responses to environmental changes and their diversity would be key to predicting the community-level responses under climate change.

## Conclusions

The present study demonstrated that the strengths of fish species interactions changed with water temperature under field conditions. For several fish species, species interactions were intensified in warmer water and for some other fish species, species interactions were weakened in warmer water. This may change the correlation and distribution of interaction strengths in a community, which may consequently influence community dynamics and stability in a complex way (*Allesina et al., 2015*; *Tang et al., 2014*; *Ushio et al., 2018a*). Therefore, a more detailed understanding of the effects of environmental conditions on species interactions under field conditions will be required to improve our capability to understand, forecast, and even manage natural ecological communities and dynamics, which is particularly important under ongoing global climate change. Developments and improvements in eDNA techniques and statistical analyses are still required; however, because eDNA analysis is potentially applicable to any type of organisms even if they are difficult to be detected using traditional methods, our framework integrating an eDNA analysis and advanced statistical analyses pave a way to understand and forecast dynamics of various ecological communities under field conditions.

## Acknowledgements

We thank T Komai, RO Gotoh, T Sunobe, and K Takiguchi for assisting with the bimonthly collection of eDNA samples. We thank the members of the eDNA project supported by JST CREST, who contributed to discussions regarding the idea of detecting interspecific interactions using environmental DNA time series and nonlinear time series analysis. This research was supported by JSPS KAKENHI (B) Grant Number 20H03323, the Hakubi Project in Kyoto University, and The Hong Kong University of Science and Technology Startup Fund to MU, and JSPS KAKENHI (B) Grant Numbers 19H03291, 22H02691, MEXT OGAP Project Grant Number JPMXD0618068274, and JST CREST Grant Number JPMJCR13A2 to MM.

## Additional information

### Funding

| Funder | Grant reference number | Author |
|---|---|---|
| Japan Society for the Promotion of Science | 20H03323 | Masayuki Ushio |
| Japan Society for the Promotion of Science | JP19H03291 | Masaki Miya |
| Japan Society for the Promotion of Science | 22H02691 | Masaki Miya |
| Kyoto University | Hakubi Project | Masayuki Ushio |
| Hong Kong University of Science and Technology | Startup Fund | Masayuki Ushio |
| Ministry of Education, Culture, Sports, Science and Technology | JPMXD0618068274 | Masaki Miya |
| Japan Science & Technology Agency | JPMJCR13A2 | Masaki Miya |

The funders had no role in study design, data collection, and interpretation, or the decision to submit the work for publication.

### Author contributions

Masayuki Ushio, Conceptualization, Data curation, Software, Formal analysis, Supervision, Funding acquisition, Validation, Visualization, Methodology, Writing – original draft, Writing – review and editing; Testuya Sado, Takehiko Fukuchi, Data curation, Investigation, Methodology, Writing – review and editing; Sachia Sasano, Reiji Masuda, Methodology, Writing – review and editing; Yutaka Osada, Resources, Software, Methodology, Writing – review and editing; Masaki Miya, Conceptualization, Data curation, Supervision, Funding acquisition, Validation, Investigation, Visualization, Methodology, Writing – original draft, Project administration, Writing – review and editing

### Author ORCIDs

Masayuki Ushio ![ORCID] https://orcid.org/0000-0003-4831-7181

Reviewer #1 (Public Review): https://doi.org/10.7554/eLife.85795.3.sa1
Reviewer #2 (Public Review): https://doi.org/10.7554/eLife.85795.3.sa2
Author response https://doi.org/10.7554/eLife.85795.3.sa3

## Additional files

### Supplementary files

Supplementary file 1. Supplementary information for eDNA metabarcoding and statistical analyses. (**a**) List of species detected from MiFish environmental DNA (eDNA) metabarcoding with raw numbers of reads. (**b**) A faunal inventory of the coastal marine fishes of Chiba prefecture compiled from museum collections and literature surveys. Museum acronyms are Natural History Museum and Institute, Chiba (CBM) and its coastal branch (CHMH), National Museum of Nature and Science, Tokyo (NSMT), Kanagawa Prefectural Museum of Natural History (KPM), and Yokosuka City Museum (YCM). All references can be found in the footnote. (**c**) Top 20 fish–fish interactions based on UIC and their interpretations. (**d**) Results of GAMM between interaction strengths and environmental and ecological properties.

Transparent reporting form

## Data availability

Source scripts for the analyses and figure generations are archived in Zenodo and publicly available at Github (copy archived at *Ushio, 2023*). DNA data are deposited DDBJ Sequence Read Archive (DRA submission ID = DRA014111).

The following datasets were generated:

| Author(s) | Year | Dataset title | Dataset URL | Database and Identifier |
|---|---|---|---|---|
| Ushio M | 2023 | ong8181/eDNA-BosoFish-network: v0.9.0 | https://doi.org/10.5281/zenodo.7865959 | Zenodo, 10.5281/zenodo.7865959 |
| Ushio M | 2022 | DRA014111 | https://ddbj.nig.ac.jp/resource/sra-submission/DRA014111 | DDBJ, DRA014111 |

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
