## [Editor Report · eLife assessment]

This study presents **important** findings regarding the quantification of dynamics in fish communities in changing ecosystems by combining a large-scale environmental DNA metabarcoding time series with novel statistical approaches. The methods are **convincing**, with controlled experiments, thorough statistical analyses, and a substantial dataset covering two years of detailed observation, which can provide sufficient power to detect fine-scale ecological interactions. This work is relevant for informing future research on assessing community stability under climate change.

---

## [Referee Report · Reviewer #1 (Public Review)]

The authors have studied the effect of temperature on the interspecific interaction strength of coastal marine fish communities, using eDNA samples. Their introduction describes the state of the art concerning the dynamics of interspecific interactions in ecological communities. This introduction is well written and highly information dense, summarizing all that the reader needs to know to further understand their study setup and execution.

The authors hypothesize that water temperature changes could have an effect on the interspecific interaction strength between marine fishes, and they studied this with a two year long, bi-weekly eDNA sampling campaign at 11 study sites in Japan with different temperature gradients. These 550 water samples were analysed for fish biodiversity through eDNA-metabarcoding using MiFish primers. By using the most abundant fish species as an internal spike in and quantifying the copy numbers from this species by qPCR, the authors were able estimate DNA copy numbers for the total dataset. From the 50 most frequently detected fish species in these samples they showed that temperature affected the interspecific interaction strength between some species. Their work provides a highly relevant approach to perform species-interaction strength analysis based on eDNA biodiversity assessments, and as such provides a research framework to study marine community dynamics by eDNA, which is highly relevant in the study of ecosystem dynamics. The models and analytical methods used are clearly described and made available, enabling application of these methods by anyone interested in applying it to their own site and species group of interest.

Strengths: The authors have a study setup that is suitable to measure the effects of temperature of the eDNA diversity, and have taken a large number of samples and all appropriate controls to be able to accurately measure and describe these dynamics. The applied internal spike in to enable relative eDNA copy number quantification is convincing.

Weaknesses:

The authors were able to find a correlation between water temperature and interaction strengths observed. However, since water temperature is dependent on many environmental variables that are either directly or indirectly influencing ecosystem dynamics, it is hard to prove a direct correlation between the observed changes in community dynamics and the temperature alone

---

## [Referee Report · Reviewer #2 (Public Review)]

In this work Ushio et al. combine environmental DNA metabarcoding with novel statistical approaches to demonstrate how fish communities respond to changing sea temperatures over a seasonal cycle. These findings are important due to the need for new techniques that can better measure community stability under climate change. The eDNA metabarcoding dataset of 550 water samples over two years is, I feel, of sufficient scale to provide power to detect fine-scale ecological interactions, the experiments are well controlled, and the statistical analysis is thorough.

The major strengths of the manuscript are: (1) the magnitude of the dataset, which provides densely replicated sampling that can overcome some of the noise associated with eDNA metabarcoding data and scale up the number of data points to make unique inferences; (2) the novel method of transforming the metabarcode reads using endogenous qPCR "spike-in" data from a common reference species to obtain estimates of DNA concentration across other species; and (3) the statistical analysis of time-series and network data and translating it into interaction strengths between species provides a cross-disciplinary dimension to the work.

I feel like this kind of study showcases the power of eDNA metabarcoding to answer some really interesting questions that were previously unobtainable due to the complexities and cost of such an exercise. Notwithstanding the problems associated with PCR primer bias and PCR stochasticity, the qPCR "spike-in" method is easy to implement and will likely become a standardised technique in the field. Further studies will examine and improve on it.

Overall I found the manuscript to be clear and easy to follow for the most part. I did not identify any serious weaknesses or concerns with the study, although I am not able to comment on the more complex statistical procedures such as the "unified information-theoretic causality" method devised by the authors. The section on limitations of the study is important and acknowledges some issues with interpretation that need to be explained. The methods, while brief in parts, are clear. The code used to generate the results has been made available via a GitHub repository. The figures are clear and attractive.

---

## [Author Response]

The following is the authors' response to the original reviews.

**eLife assessment**
This study presents important findings regarding the quantification of dynamics in fish communities in changing ecosystems by combining a large-scale environmental DNA metabarcoding time series with novel statistical approaches. The methods are convincing, with controlled experiments, thorough statistical analyses, and a substantial dataset covering two years of detailed observation, which can provide sufficient power to detect fine-scale ecological interactions. This work is relevant for informing future research on assessing community stability under climate change.

Thank you so much for your careful evaluation of our manuscript. We are very pleased to hear that you found our study important. We have revised our manuscript according to the helpful comments to further improve our manuscript.

**Reviewer #1 (Public Review):**
[…] Their work provides a highly relevant approach to perform species-interaction strength analysis based on eDNA biodiversity assessments, and as such provides a research framework to study marine community dynamics by eDNA, which is highly relevant in the study of ecosystem dynamics. The models and analytical methods used are clearly described and made available, enabling application of these methods by anyone interested in applying it to their own site and species group of interest.

Thank you so much for your time and effort to evaluate our manuscript. We are very pleased to hear that you found our study interesting. We have further revised the manuscript according to your comments and hope that the revised manuscript is now better than the original one.

Strengths: The authors have a study setup that is suitable to measure the effects of temperature of the eDNA diversity, and have taken a large number of samples and all appropriate controls to be able to accurately measure and describe these dynamics. The applied internal spike in to enable relative eDNA copy number quantification is convincing.

We are happy to hear that you found the study design and the method to estimate eDNA copy number are suitable and convincing.

Weaknesses: The authors aim to study the relationship between species interaction strength and ecosystem complexity, and how temperature will influence this. However, there is only limited ecological context discussed explaining their results, and a link with climate change scenario's is also limited. A further discussion of this would have strengthened the manuscript.

Thank you so much for the comment. We have added discussion about how our study contributes to understanding fish community assembly process and predicting the community-level response under ongoing climate change. We have added one subsection, *"Implications for fish community assembly and the effect of global climate change* ", at **L679**. As for the ecological discussion for each specific fish-fish interaction, we provided this in Supplementary file 1c.

The authors were able to find a correlation between water temperature and interaction strengths observed. However, since water temperature is dependent on many environmental variables that are either directly or indirectly influencing ecosystem dynamics, it is hard to prove a direct correlation between the observed changes in community dynamics and the temperature alone.

Thank you for pointing this. We have discussed the possibility of the effects of other environmental variables (e.g., oxygen) and how we could overcome this issue at **L661**. Some of the sentences were originally in the subsection " *Interaction strengths and environmental variables* ", but were moved to the subsection " *Potential limitations of the present study and future perspectives*".

**Reviewer #2 (Public Review):**
In this work Ushio et al. combine environmental DNA metabarcoding with novel statistical approaches to demonstrate how fish communities respond to changing sea temperatures over a seasonal cycle. These findings are important due to the need for new techniques that can better measure community stability under climate change. The eDNA metabarcoding dataset of 550 water samples over two years is, I feel, of sufficient scale to provide power to detect fine-scale ecological interactions, the experiments are well controlled, and the statistical analysis is thorough.

Thank you so much for your time and effort to evaluate our manuscript. We are happy to hear that you found our study technically sound and important. We have revised the manuscript according to your comments to improve our manuscript further.

The major strengths of the manuscript are: (1) the magnitude of the dataset, which provides densely replicated sampling that can overcome some of the noise associated with eDNA metabarcoding data and scale up the number of data points to make unique inferences; (2) the novel method of transforming the metabarcode reads using endogenous qPCR "spike-in" data from a common reference species to obtain estimates of DNA concentration across other species; and (3) the statistical analysis of time-series and network data and translating it into interaction strengths between species provides a cross-disciplinary dimension to the work.

Thank you for your positive comments. Regarding (1), we are very pleased to hear that (1) our intensive and extensive water sampling, (2) our method for using the common fish species eDNA as "spike-in," and (3) our nonlinear time series analysis were positively evaluated.

I feel like this kind of study showcases the power of eDNA metabarcoding to answer some really interesting questions that were previously unobtainable due to the complexities and cost of such an exercise. Notwithstanding the problems associated with PCR primer bias and PCR stochasticity, the qPCR "spike-in" method is easy to implement and will likely become a standardised technique in the field. Further studies will examine and improve on it.

We must admit that our endogeneous "spike-in" method does not overcome all problems associated with PCR. However, we agree with you and believe that we are heading in a correct direction. The method

does not require the addition of external internal standard DNAs and enables post-hoc evaluation of eDNA absolute concentrations. Although this approach requires an additional experiment (qPCR), the method may be an alternative for quantifying eDNA concentrations.

Overall I found the manuscript to be clear and easy to follow for the most part. I did not identify any serious weaknesses or concerns with the study, although I am not able to comment on the more complex statistical procedures such as the "unified information-theoretic causality" method devised by the authors. The section on limitations of the study is important and acknowledges some issues with interpretation that need to be explained. The methods, while brief in parts, are clear. The code used to generate the results has been made available via a GitHub repository. The figures are clear and attractive.

We are very happy to hear that you found our manuscript clear and not containing any serious weakness.

**Reviewer #1 (Recommendations For The Authors):**
This is a very nice manuscript discussing highly relevant methods to use eDNA analysis to study interactions in marine ecosystems. There are some minor concerns that we will address below:- As already mentioned above, based on the statements in the introduction we expected a very elaborate discussion section concerning the ecological interaction observed between species. This is however missing, and a more extensive general discussion of the biological interactions would be appreciated, either based on existing literature, or by suggesting further experiments. Alternatively, the claims made in e.g. line 124-128 (Overcoming these difficulties....) could be amended so this expectation is not raised.

Thank you so much for the comment. As answered in the response above, we have added discussion about how our study contributes to the fish community assembly process and predicting the community-level response under ongoing climate change at **L679**.

Specifically, we argued that our study provides a piece of evidence that temperature exerts influences on fish-fish interactions under field conditions at a relatively short time scale (weeks to months). We suggested that temperature effects on fish community assembly involve effects at different time scales, and thus, integrating results from different temporal (and spatial) scales are necessary to understand the fish community assembly process in nature. As stated above, we provided the detailed ecological discussion for each specific fish-fish interaction in the Supporting Information.

- A lot of negative controls were taken and described in the material & methods. However, there is no clear mention of what was done with the outcome of these negative controls. How did the results of the negative controls influence your analysis? Or were they all completely negative?

Thank you for pointing this out. The negative controls produced negligible reads (177 ± 665 reads [mean ± S.D.]), which accounted for ca. 0.1% of the positive sample reads. Moreover, all the reads were assigned to non-target taxa, such as fish species that had never been observed in the study region and freshwater fish species. Therefore, we conclude that any contaminations in our experiments were negligible, and we discarded the sequence reads from the negative control samples. We have explained this in **L533–L539** in the main text.

- Line 423 states: "..suggesting that weak interactions are key to the maintenance of species-rich communities." We are wondering if this can be stated like this, as it seems the other way around would also be true, since in a species rich community it can be expected that most interactions are weak?

Thank you for pointing this. out We agree that there is a possibility that the high species diversity could be a cause of weak intearctions. To clarify this, we have revised the sentence as follows in **L568**: " *...suggesting that understanding the causes and effects of weak interactions is key to understanding the maintenance of species-rich communities.* "

- There is a correlation between DNA concentration and temperature (e.g. shown in fig. S2b). We wondering what could be an argument to not correct for this temperature effect on eDNA concentrations (as now described) or if it would be better to apply a correction factor for this, as it is also shown that there is a correlation between DNA concentration and interaction strengths.

In the unified information theoretic (UIC) analysis, we took the effect of temperature into account if temperature had statistically clear influence on eDNA dynamics of a particular fish species (**L439**). This means that temperature was included as a conditional variable in the calculation of TE (i.e., *Zt* in Eqn. [1]). Other environmental variables were also included if they had statistically clear influence. Similarly, in the MDR S-map, we included temperature or other environmental variables as conditional variables if they had statistically clear influence on eDNA dynamics of a particular fish species. We explained this in **L479**.

- The models used for the interaction dynamics calculations are extensively discussed in this manuscript, although these details are also present in the original papers describing these models, and therefore the manuscript could be shortened by removing some of this explanation.

Thank you for your suggestion. As you understood, the details of the method (S-map and MDR S-map) are available in Sugihara (1994), Chang et al. (2021), and elsewhere. However, we have kept the explanation so that readers who are not familiar with the methods can briefly understand the methods without the needs to read the detail of the previoius studies.

**Reviewer #2 (Recommendations For The Authors):**
L50-L72: I feel like the abstract could be snappier, i.e. quicker to read with less detail. Consider reducing it a little.

Thank you for your suggestion. We have deleted some redundant phrases and shortened the abstract a little.

L173-L176: I don't understand exactly what is suggested here. Perhaps rephrase?

We have revised the sentence as follows (**L165**): " *As our eDNA time series was taken twice a month, the interactions detected should also have the same time scale (e.g., the interactions detected may cause changes in the population size at the same time scale), which means that we tend to focus on behavior-level interactions (e.g., schooling) rather than birth-death process in the present study (except for predation).*"

L228: How many PCR replicate reactions were undertaken per sample?

We performed eight technical replicates for the same eDNA template. This information is described in the third paragraph of the section "Paired-end library preparation and MiSeq sequencing." This section has been moved from the previous supplementary methods to the main text in the revision.

L236: There is no mention later of how these blanks are used to clean up or filter the dataset from the effects of contamination. Consider adding this information.

Thank you for pointing this. As in the responses above, we have described the negative controls in **L533–L539** in the main text. The negative controls generated negligible reads, so we simply discarded the sequence reads.

L252-L253: "Primer sequences were removed from merged reads and reads without the primer sequences underwent quality filtering"? Wouldn't all of the reads not have primers after the primers were trimmed off? Or is something else intended here?

All primer sequences were removed after merging the paired- end reads (see "Sequence analysis"). There is no specific reason for this process, and we think that the primer removal before merging the paired- end reads will generate the same results.

L264-L265: "To refine the above taxon assignments". I assume because there were lots of assignments to species that were not known from the study area? Explain why this was done.

At present, the reference sequences are available for about 70% of 4,500 fish species in Japan. However, due to the unknown degree of intraspecific variation, using a uniform threshold of 98.5% to delineate species can result in over-splitting or over-clustering MOTUs. To solve this issue, the manual refinement of the taxon assignments was performed based on the phylogenetic tree. This has been explained in **L335**.

L274: More details of the qPCR assay are required, or a citation of previous study or supporting information.

The details of the qPCR assay are provided in the secion "Quantitative PCR and estimation of DNA copy numbers." This section has been moved from the previous supplementary methods to the main text in the revision.

L327: Explain further how seasonality was treated here? This is an important part of the study, so deserves further attention.

We included water temperature (if it had statistically clear influence on fish eDNA dynamics) as a conditional variable *z(t*) in the calculation of TE, and this took the effect of the seasonality in detecting causation into account. We have described this in **L436–444**.

L407: Consider giving the code repository a DOI to cite.

We have archived the analysis codes at Zenodo and provided the DOI in **L39** and **L521**.

L411: How many MiSeq runs exactly?

We performed 21 MiSeq runs (often with other eDNA samples). We have described this in the main text (**L299**).

L411: What proportion of your total sequencing data were assigned to fishes? This is a useful statistic to compare methods between studies.

About 98% of the total sequence reads was assigned to fish. We have described this in the main text (**L528**).

Figure 2: There does not appear to be a key to the color-coded species ecologies.

We have added a legend for the fish ecology in Figure 2.